# How do midwives facilitate women to give birth during physiological second stage of labour? A systematic review

Maria Healy[1]*, Viola Nyman[2,3], Dale Spence[1], René H. J. Otten[4], Corine J. Verhoeven[5,6,7]

1 School of Nursing and Midwifery, Queen's University Belfast, Belfast, Northern Ireland, United Kingdom, 2 Department of Research and Development, NU-Hospital Group, Trollhattan, Sweden, 3 Institute of Health and Care Sciences, University of Gothenburg, Gothenburg, Sweden, 4 University Library, Vrije Universiteit Amsterdam, Amsterdam, Netherlands, 5 Department of Midwifery Science, AVAG, Amsterdam Public Health Research Institute, Amsterdam UMC, VU Medical Centre, Amsterdam, Netherlands, 6 Department of Obstetrics and Gynaecology, Maxima Medical Centre, Veldhoven, Netherlands, 7 Division of Midwifery, School of Health Sciences, University of Nottingham, Nottingham, United Kingdom

* maria.healy@qub.ac.uk

**Data Availability Statement:** All relevant data are within the paper and its Supporting Information files.

## Abstract

Both nationally and internationally, midwives' practices during the second stage of labour vary. A midwife's practice can be influenced by education and cultural practices but ultimately it should be informed by up-to-date scientific evidence. We conducted a systematic review of the literature to retrieve evidence that supports high quality intrapartum care during the second stage of labour. A systematic literature search was performed to September 2019 in collaboration with a medical information specialist. Bibliographic databases searched included: PubMed, EMBASE, Cumulative Index to Nursing and Allied Health Literature (CINAHL), PsycINFO, Maternity and Infant Care Database and The Cochrane Library, resulting in 6,382 references to be screened after duplicates were removed. Articles were then assessed for quality by two independent researchers and data extracted. 17 studies focusing on midwives' practices during physiological second stage of labour were included. Two studies surveyed midwives regarding their practice and one study utilising focus groups explored how midwives facilitate women's birthing positions, while another focus group study explored expert midwives' views of their practice of preserving an intact perineum during physiological birth. The remainder of the included studies were primarily intervention studies, highlighting aspects of midwifery practice during the second stage of labour. The empirical findings were synthesised into four main themes namely: birthing positions, non-pharmacological pain relief, pushing techniques and optimising perineal outcomes; the results were outlined and discussed. By implementing this evidence midwives may enable women during the second stage of labour to optimise physiological processes to give birth. There is, however, a dearth of evidence relating to midwives' practice, which provides a positive experience for women during the second stage of labour. Perhaps this is because not all midwives' practices during the second stage of labour are researched and documented. This systematic review provides a valuable insight of the empirical evidence relating to midwifery practice during the physiological second stage of labour, which can also inform

**Funding:** This article is based upon work funded by the COST Action IS1405 BIRTH: "Building Intrapartum Research Through Health - An interdisciplinary whole system approach to understanding and contextualising physiological labour and birth" (http://www.cost.eu/COST_Actions/isch/IS1405), supported by EU COST (European Cooperation in Science and Technology). Furthermore, the School of Nursing and Midwifery, Queen's University Belfast, funded access to Covidence, the web-based systematic review software package recommended by Cochrane. The funders had no role in study design, data collection and analysis, decision to publish, or preparation of the manuscript.

**Competing interests:** The authors have declared that no competing interests exist.

**Abbreviations:** CERQual, Confidence in the Evidence from Reviews of Qualitative research; GRADE, Grading of Recommendations Assessment, Development and Evaluation; PEO, Population, Exposure, Outcomes; PICO, Patient or Population, Intervention, Comparison, Outcome; PRISMA, Preferred Reporting Items for Systematic Reviews and Meta-Analysis; PROSPERO, International Prospective Register of Systematic Reviews; WHO, World Health Organization.

education and future research. The majority of the authors were members of the EU COST Action IS1405: Building Intrapartum Research Through Health (BIRTH). The study protocol is registered in the International Prospective Register of Systematic Reviews (PROSPERO; Registration CRD42018088300) and is published (Verhoeven, Spence, Nyman, Otten, Healy, 2019).

## Introduction

Childbirth is a significant and memorable life event for a woman and her family. Women's experiences of birth have both short and long-term effects on their health and wellbeing for both themselves and their infants [1–6]. As stated by the World Health Organization (WHO) in 2018, the primary outcome for all pregnant women is to have a '*positive childbirth experience*'. This includes giving birth to a healthy baby in a conducive, safe environment with continuity of care provided by kind, competent maternity care professionals [7]. In addition, the WHO has highlighted that most women value a physiological labour and birth. Experiencing physiological childbirth also has a long-term impact: '*The health and well-being of a mother and child at birth largely determines the future health and wellness of the entire family*' [8]. Furthermore, childbirth has physical effects on women and their future pregnancies. Although cesarean delivery is associated with a reduced rate of urinary incontinence and pelvic organ prolapse, it is also associated with increased risks for fertility, future pregnancy, and long-term childhood outcomes such as increased odds of asthma and obesity [9].

Normal physiological birth was defined by the WHO as '*spontaneous in onset, low-risk at the start of labour and remaining so throughout labour and delivery. The infant is born spontaneously in the vertex position between 37 and 42 completed weeks of pregnancy. After birth mother and infant are in good condition*' [10]. Labour can be divided into three stages: the first, second and third stage of labour. The first stage of labour is defined as the time period characterised by regular painful uterine contractions until full dilatation of the cervix and the second stage of labour as the time period between full dilatation of the cervix and the birth of the baby, whilst the woman is experiencing an involuntary urge to bear down, due to expulsive uterine contractions [7]. The third stage is recognised as the period after the birth of the baby ending with the birth of the placenta and fetal membranes [11].

Normal physiological birth is associated with the non-use of an epidural or other pharmacological pain relief, as it may affect the natural course of labour and can lead to rare but potentially severe adverse maternal effects [10, 12]. The same accounts for induction and augmentation of labour. Especially high doses of synthetic oxytocin may cause more and longer painful contractions when compared to normal labour [13]. Uvnäs-Moberg has highlighted how the process of physiological labour and birth can be enabled by the interplay of reproductive hormonal and neuro-hormonal mechanisms when the midwife provides kind and respectful caring practices. These practices promote oxytocin release for effective uterine contractions during labour and the relaxation of the birth canal [14, 15]. Little is known of the variety of physical and emotional actions the midwife does when '*being with*' a woman during birth of the baby, in particular, how midwives facilitate this physiological process. According to Kennedy et al. it is a research priority to identify and highlight aspects of care that optimise, and those that disturb, the biological/physiological processes during childbirth [16].

The objective of this systematic review was therefore, to examine the evidence relating to intrapartum midwifery care, focusing specifically on care during the second stage of labour.

The structured research questions were formulated using the PICO (Patient or Population, Intervention, Comparison, Outcome) framework for quantitative research and the PEO (Population, Exposure, Outcomes) question format for qualitative research questions: *'How do midwives facilitate women to give birth during physiological second stage of labour*?

The results of this systematic review will support high quality intrapartum care during the second stage and inform midwifery practice, education and future research and positively influence this aspect of midwifery care for women.

## Methods

We undertook a systematic literature search based on the Preferred Reporting Items for Systematic Reviews and Meta-Analysis (PRISMA) statement (S1 Checklist) [17]. The Peer Review of Electronic Search Strategies (PRESS) 2015 Guideline Statement was used to enhance the quality and comprehensiveness of the electronic literature search [18]. The PICO framework for quantitative and PEO framework for qualitative studies were also utilised: P: women in second stage of labour, I: intrapartum intervention by midwives, C: standard care, O: spontaneous physiological birth. PEO framework: P: women in second stage of labour, E: midwives' practices in the second stage of labour, O: spontaneous physiological birth. Systematic searches of the bibliographic databases: EMBASE.com, Cinahl, PsycINFO, PubMed, Maternity and Infant Care Database and The Cochrane Library were conducted.

The search strategy included the Boolean terms OR and AND, the search terms included controlled terms (for example, MeSH terms in PubMed and Emtree in Embase) as well as free text terms and truncations (*) (S1 Table). We used free text terms only in The Cochrane Library and synonyms and variations of the keywords in all databases. The search terms include: "Labor, Obstetric"[Mesh] OR "Parturition"[Mesh] OR "Delivery, Obstetric" [Mesh] OR labor [tiab] OR labour[tiab] OR birth*[tiab] OR childbirth*[tiab] OR parturition*[tiab] OR deliver*[tiab] OR "Labor, Stage, Second"[Mesh], see Fig 1.

### Inclusion/exclusion criteria

Only full text articles published in peer-reviewed journals were included. All languages were accepted, as the authors were part of the EU COST Action IS1405: *Building Intrapartum Research Through Health (BIRTH)* network and therefore had access for most languages to be translated, if necessary. All studies describing midwives' care or practice during second stage of physiological birth or normal birth were included. Both relevant quantitative and qualitative studies were eligible for review.

Case studies were excluded. Studies examining midwifery practice of women that focused only on care during the first or third stage of labour were excluded. Studies including women who had an epidural, spinal, operative vaginal birth or caesarean section were also excluded. Furthermore, studies that included women, who had a preterm birth, had their pregnancy induced or labour augmented with intravenous oxytocin were excluded. Searches of the bibliographic databases were undertaken initially from inception to 8th May 2018. The search was further refined to include papers published from 1st January 2008 to 8th May 2018, reflecting the National Institute for Health and Care Excellence (NICE) [19] Intrapartum care guidance which was updated at the end of 2007. Furthermore, we updated the search to 5th September 2019, in collaboration with a medical librarian. Animal studies were excluded.

Studies were selected for inclusion following a two-stage process using *Covidence*, which is a web-based software platform that streamlines the production of systematic reviews, including Cochrane reviews. Within the first screening stage each study had the title and abstract screened by pairs of two independent reviewers (CV, DS, VN, MH) and studies were excluded

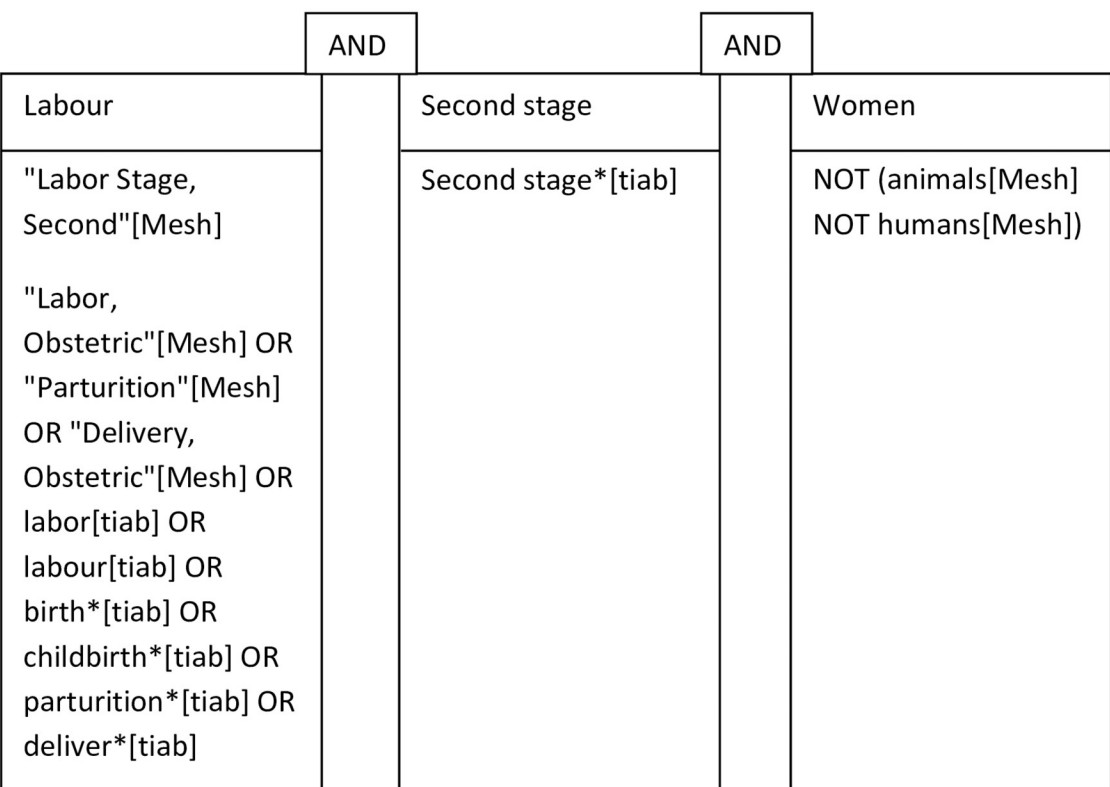

**Fig 1. Search strategy.**

if both reviewers considered a study did not meet the eligibility criteria. Full text manuscripts of the selected studies were then retrieved. Two reviewers independently, made the final inclusion or exclusion decisions on examination of the full text manuscripts. Any disagreements were discussed and resolved by a lead review author (MH or CV). The reasons for study exclusion were reported in the PRISMA flow diagram, see Fig 2.

## Quality assessment

Articles that passed the two-stage screening process then underwent quality assessment and their reference lists were hand searched. The tools utilised to assess the quality of evidence depended on each study's methodological approach. To assess the risk of bias in randomised controlled trials the Cochrane Collaboration's tool for assessing risk of bias was used [20] (Table 1). For all other study designs the Critical Appraisal Skills Programme (CASP) criteria was used (Critical Appraisal Skills Programme 2018) [21]. The Grading of Recommendations Assessment, Development and Evaluation (GRADE), the Cochrane's recommended approach for grading the body of evidence, was also utilised for the quantitative studies. Confidence in the Evidence from Reviews of Qualitative research (CERQual) was used for grading the confidence in the evidence of qualitative studies.

## Results

The systematic search resulted in 13,034 records initially imported into Mendeley (a reference manager) aiding detection of duplicates, leaving 7,108 imported for screening into Covidence.

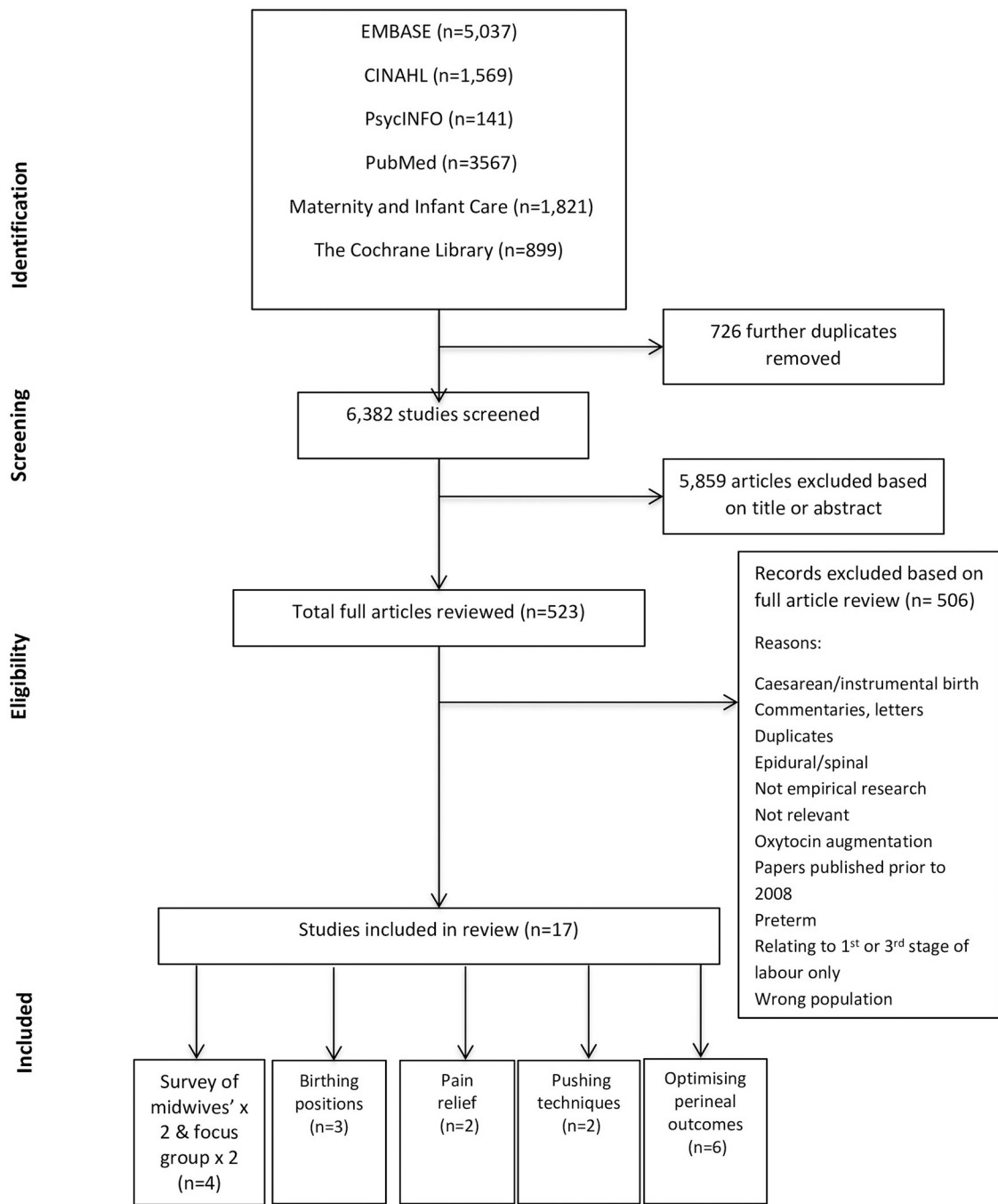

**Fig 2. Preferred Reporting Items for Systematic Reviews and Meta-Analysis (PRISMA) flow chart of articles included.**

Further duplicates were detected by Covidence, with 6,382 remaining for screening. Titles and abstracts were then reviewed; subsequently 523 articles were retrieved for full text assessment. Following detailed review 506 articles did not meet the inclusion criteria leaving 17 studies included in this systematic review. Fig 2 summarises the search strategy and the reasons for exclusion. Studies were grouped according to the study subject and for each study a data extraction matrix was completed. The matrix comprised of ten key features of the study

**Table 1. Risk of bias.**

| Studies Name et al, Year | Random sequence allocation (selection bias) | Allocation concealment (selection bias) | Blinding of participants & personnel (performance bias) | Blinding of outcome (detection bias) | Incomplete outcome data (attrition bias) | Selective reporting (reporting bias) | Other bias |
|---|---|---|---|---|---|---|---|
| Alihosseni *et al.* (2018) | 🟥 | 🟨 | 🟥 | 🟨 | 🟩 | 🟨 | 🟨 |
| Fahami *et al.* (2011) | 🟨 | 🟩 | 🟥 | 🟨 | 🟩 | 🟩 | 🟥 |
| Shahoei *et al.* (2017a) | 🟨 | 🟩 | 🟥 | 🟨 | 🟩 | 🟩 | 🟥 |
| Shahoei *et al.* (2017b) | 🟨 | 🟩 | 🟥 | 🟨 | 🟩 | 🟩 | 🟥 |
| Valiani *et al.* (2016) | 🟨 | 🟨 | 🟥 | 🟥 | 🟩 | 🟨 | 🟨 |
| Vaziri *et al.* (2016) | 🟨 | 🟨 | 🟥 | 🟨 | 🟨 | 🟩 | 🟨 |

Red = High

Yellow = Unclear

Green = Low

including: theme, author, year, country, study design, quality assessment, relevant participant data, outcomes assessed, summary of the findings, comments (Table 2).

The seventeen included publications dated from 2008 to 2019. The majority of the studies were systematic reviews (n = 6, of which 3 were Cochrane reviews) [22–27], randomised controlled trials (n = 6) [28–33], one cohort studies with prospective data collection [34], two surveys [35, 36] and two qualitative focus group studies [37, 38].

The methodological quality of the 17 included studies was assessed. Table 1 shows the risk of bias in randomised controlled trials [28–33]. Most studies were of low or moderate quality, only the systematic reviews were of high quality [22–27]. The cohort study was assessed by CASP as good quality [34], both surveys were assessed as being of moderate quality [35, 36]. Quality assessment of the qualitative studies was assessed by Cerqual, resulting in a moderate level of confidence [37, 38].

Two studies emerged from the literature having surveyed midwives regarding their practice in the second stage of labour. One explored 1,496 midwives' practices in France, throughout the second stage of labour [35], while the other focused on 607 midwives' practices in England regarding '*hands on or hands off*' the perineum at birth [36]. The Barasinski *et al.*, [35] study highlighted that midwives' practices were influenced by their years of experience and the designation of the maternity unit where they worked [35]. The units ranged from Level I to Level III (Level I = maternity ward without a neonatology department for women with straightforward pregnancy, Level III = maternity ward with a neonatology department and neonatal intensive care unit). The survey found that the practices reported by the midwives in France were not always consistent with the scientific literature and that they could not always ensure the physiological approach to birth; particularly the midwives working in the level III units. This was in comparison to midwives working in the level I units, where women were most often encouraged to adopt non-horizontal positions, could choose which method of pushing they preferred (valsalva or open glottis pushing) and significantly, an increased number of midwives in these units reported using warm compresses on the perineum during the second stage of labour. The survey of midwives in England [36] found that 299 (49.3%, 95% CI 45.2–53.3%) midwives preferred the "hands-off" method while 48.6% preferred "hands on".

**Table 2. Data extraction matrix.**

| Midwives' practices | Author, year Country Aim of the study | Study design | Population Group and size (n) (age, parity, ethnicity, etc.) | Quality of study (CASP, Cerqual and GRADE) ⊕⊕⊕⊕ High ⊕⊕⊕○ Moderate ⊕⊕○○ Low ⊕○○○ Very Low | Definitions Main components | Outcomes assessed Effects on outcomes Components associated with outcomes | Results | Key conclusions | Comments |
|---|---|---|---|---|---|---|---|---|---|
| **Surveys and focus groups of midwives' practices in the second stage of labour** | | | | | | | | | |
| Birth position, pushing methods, perineal protection, perineal support techniques | **Barasinski et al., 2018** France To describe the practices reported by French midwives during the active second stage of labor | Cross-sectional Internet Survey | 1896 Midwives from 377 maternity units (hospital-based) who attended at least 1 birth in 2013. Level 1 = maternity ward without a neonatology dept. (low risk) Level 2 = maternity ward with a neonatology dept. Level 3 = maternity ward with a neonatology dept. and neonatal intensive care unit (NICU) | ⊕⊕○○ | Variety of birth positions, pushing methods (if woman led, Valsalva, Open-glottis or both), perineal protection, perineal support techniques, (perineal massage, lubricant, warm compresses, management of fetal head, Ritgen's manoeuver, restitution, delivery of head) midwives' feelings about facilitating physiological birth | | One third of midwives let women choose the type of pushing. Half of the midwives (53.5%) didn't use perineal massage. 24% of all midwives used warm compresses on the perineum with significantly more use (33.6%—P<0.0001) in level 1 units. Most midwives (91.4%) preferred the hands-on technique. 81.9% of midwives thought their labor management often/always ensured physiological birth. Overall, only 38.2% of midwives were comfortable with all the maternal positions for birth. | Practices reported by French midwives are not always consistent with the scientific literature or with a physiological approach to birth. These practices vary based on experience and type of unit where they work. There is an absence of professional guidelines for midwives in France. | |
| A qualitative exploration of techniques used by expert midwives to preserve the perineum intact | **Begley et al., 2019** Ireland and New Zealand To explore expert Irish and New Zealand midwives' views of the skills that they employ in preserving the perineum intact during spontaneous vaginal birth. | Focus Group Expert midwives (from New Zealand and Ireland) | 21 midwives with 7 from Ireland and 14 from New Zealand -Mean length of time working as a midwife was 16.6 years (SD 10.6), range 5–36 years. | Moderate level of confidence | Expert was defined as achieving, in the preceding 3.5 years, an episiotomy rate for nulliparous women of less than 11.8% (the mean rate from all NZ and Irish MLU data combined), a 'no suture' rate (combination of first degree tears did not require sutures, and intact perineums of more than 40%, and a rate of less than 3.2% for serious perineal tears (or one third/ fourth degree tear) | | Four core themes were identified from the data on participants' expertise in relation to techniques they used during birth to preserve the perineum. These were: '*Calm, controlled birth*', which involved: Developing an empowering, trusting relationship with the woman, ensuring a quiet, calm environment and preparing, reassuring and supporting the woman' '*Position and techniques in early second stage*' involving: Encouraging women to use upright positions or those with free movement of the sacrum (lateral, leaning back from a birthing stool, on all fours), using hot compresses on the perineum, Consider gel or oil for lubrication. '*Hands on or off?*' If the woman is in control, then there is potential for hands closely poised in preparation to apply pressure if head advances rapidly. Use gentle pressure on the head to control explosive forces, supporting either side of the perineum with the index finger and thumb, with the second hand palm holding pad over the anus. 'Easing' the two sides together to create some slack in the perineum, if necessary. '*Slow, blow and breathe the baby out.* Ensure slow birth of the head while the woman breathes or blows, providing enough time during crowning to allow the perineum to stretch slowly and fully (5 or more contractions, if the fetal heart is satisfactory), waiting for shoulders to rotate, and ease up off the perineum. | Four core themes were identified: 'Calm, controlled birth'; 'Position and techniques in early second stage'; 'Hands on or off?' and 'Slow, blow and breathe the baby out. Using the techniques described enabled these midwives to achieve rates, in nulliparous women, of 3.91% for episiotomy, 59.24% for 'no sutures', and 1.08% for serious lacerations | |
| 1000 midwives survey-routine hands on | **Trochez et al., 2011** UK To determine current midwifery practice in England with regard to the management of the perineum during the late second stage of labour | Observational postal questionnaire | Response rate 60.7% (n = 607) | ⊕⊕○○ | To estimate the number of midwives practicing either "hands on" or "hands off" | Informed consent/ choice Factors related to giving informed consent Working conditions Obstetric factors | 299 (49.3%, 95% CI 45.2–53.3%) midwives preferred the "hands-off" method; 48.6% preferred "hands on". Less-experienced midwives were more likely to prefer the "hands off" (72% vs. 41.4%, p<0.001). A higher proportion of midwives in the "hands-off" group would never do an episiotomy (37.1% vs. 24.4%, p = 0.001) for indications other than fetal distress. | Midwives in the UK apply both methods of hands on and hands off the perineum during the second stage of labour. | |
| Women's positions during second stage of labour | **De Jonge et al., 2008** Netherlands To explore the views of midwives on women's positions during second stage of labour | Qualitative focus groups | 6 focus groups with purposive sample of 31 independent primary care midwives from rural, semi-urban and urban areas from different parts of country of various ages and educational backgrounds | Moderate level of confidence | Topic guide: midwives' experience with birthing positions, info they give to women about positions, factors that influence use of positions and knowledge and skills is assisting births in various positions | | Most use birthing stool though risk of oedema Quarter stated all of last 10 births were in supine position. Giving women informed choice may assist them in using positions that are most appropriate. Midwives emphasised women should be prepared that the process of birth is largely unpredictable. Equipment for non-supine births should be more midwife friendly. | Influence of midwives' working conditions on use of birthing positions was important factor. | |

*(Continued)*

Table 2. (Continued)

| Midwives' practices | Author, year Country Aim of the study | Study design | Population Group and size (n) (age, parity, ethnicity, etc.) | Quality of study (CASP, Cerqual and GRADE) ⊕⊕⊕⊕ High ⊕⊕⊕o Moderate ⊕⊕oo Low ⊕ooo Very Low | Definitions Main components | Outcomes assessed Effects on outcomes Components associated with outcomes | Results | Key conclusions | Comments |
|---|---|---|---|---|---|---|---|---|---|
| **Birthing positions** | | | | | | | | | |
| Birthing position and perineal damage | De Jonge et al., 2010 Netherlands To examine the association between semi-sitting and sitting compared with recumbent position at the time of birth and perineal damage | Cohort (secondary analysis of RCT; not randomised for different positions) | Low risk women, 18+ (n = 1646) 640 primiparous (39.3%) 987 multips (60.7%) Mean age not given | ⊕⊕⊕o | Semi sitting was defined as sitting on a bed or birthing stool; recumbent was defined as supine or lateral. | Perineal damage Overall episiotomy rate 22.7%; 43.8% 1st or 2nd degree tear; 1.9% 3rd degree tear, 9.3% labial tear. | 922 women gave birth in recumbent position, 605 semi-sitting, 119 sitting. Primiparous women had higher chance for episiotomy [OR 1.99 (1.47–2.71)] or labial tears (OR 2.44 (1.59–3.74)). Women in sitting position were less likely to have an episiotomy [OR 0.29 (0.16–0.54)] and more likely to have a perineal tear [OR1.83 (1.22–2.73)]; Longer duration of 2nd stage was associated with more episiotomies [OR 8.02 (4.97–12.95)]; No difference in intact perineum rates between position groups. | (Semi-)sitting birthing position does not need to be discouraged to prevent perineal damage. | Larger studies needed to examine differences in OASIS between different position groups |
| Women's Position in the second stage | Gupta et al., 2017 To determine the possible benefits and risks of the use of different birth positions during the second stage of labour without epidural anesthesia, on maternal, fetal, neonatal and caregiver outcomes. | Cochrane Review | 30 trials involving 9015 women | ⊕⊕⊕⊕ | Any upright position assumed by pregnant women during the second stage of labour compared with supine or lithotomy positions. | Duration of second stage of labour. Secondary outcomes: Maternal outcomes: Pain, use of any analgesia, mode of birth, perineal trauma, episiotomy, blood loss, need for blood transfusion, manual removal of placenta, shoulder dystocia, urinary or faecal incontinence. Fetal outcomes: Abnormal fetal heart rate patterns, admission to NICU, perinatal death. | The upright position was associated with a reduction in duration of second stage in the upright group (MD -6.16 minutes, 95% CI -9.74 to -2.59 minutes); no clear difference in the rates of caesarean section (RR 1.22, 95% CI 0.81 to 1.81); a reduction in assisted deliveries (RR 0.75, 95% CI 0.66 to 0.86) and episiotomies (RR 0.75, 95% CI 0.61 to 0.92); a possible increase in second degree perineal tears (RR 1.20, 95% CI 1.00 to 1.44); no clear difference in the number of 3rd or 4th degree perineal tears (RR 0.72, 95% CI 0.32 to 1.65); increased estimated blood loss greater than 500 mL (RR 1.48, 95% CI 1.10 to 1.98); fewer abnormal fetal heart rate patterns (RR 0.46, 95% CI 0.22 to 0.93); no clear difference in the number of babies admitted to NICU (RR 0.79, 95% CI 0.51 to 1.21) | The findings of this review suggest several possible benefits for upright posture in women without epidural anesthesia, such as a very small reduction in the duration of second stage of labour (mainly from the primigravid group), reduction in episiotomy rates and assisted deliveries. However, there is an increased risk blood loss greater than 500 mL and there may be an increased risk of second-degree tears, though this is unclear. In view of the variable risk of bias of the trials reviewed, further trials using well-designed protocols are needed to ascertain the true benefits and risks of various birth positions. | The overall applicability of the upright position to reduce the duration of second stage labour should be interpreted with caution. These measures can fit into the context of current practice, especially with regard to informing women of these risks during the counselling process. |
| Three delivery positions on pain intensity during the second stage of labour | Valiani et al., 2016 Iran To investigate and compare the severity of delivery pain through different childbirth positions in the second stage of delivery. | Clinical trial (randomisation procedure not described) | 96 primiparous women; mean age 22.31 (SD 2.97); gestational age between 37 and 42 weeks; singleton pregnancy; vertex position | ⊕ooo | lithotomy: the mother was in supine position with 30˚ head elevation and bent knees. Sitting position: mother sat on the labor chair with completely straight lumbar spine, hip and knee joints at the same level. Squatting position: mother was sitting on her feet so that her sole was in touch with the floor and the knee joints were higher than the hips. | Pain severity measured by VAS and McGill PPI | In the latent phase of 2nd stage of labor, pain severity based on VAS and McGill was significantly less in squatting and lithotomy groups compared to sitting position (P = 0.001). In the active phase of second labor stage, pain severity based on VAS and McGill was less in the squatting group compared to sitting and lithotomy positions (P = 0.024). | Squatting is viewed as an easy, applicable method to reduce pain 2nd stage labour. Results suggest that the use of squatting position decreases pain severity in the second stage of labor. It is also suggested to educate the mothers concerning all childbirth positions and let them select each of the positions voluntarily. | Further studies can clarify the advantages and disadvantages of all positions. |
| **Non-pharmacological pain relief** | | | | | | | | | |
| Heat therapy on pain severity | Fahami et al., 2011 Iran To assess the effect of heat therapy on pain severity in primigravida women | RCT | N = 64 Low risk nulliparous women 18–35 yrs 37–41 weeks, single pregnancy, cephalic presentation. Risk of selection bias: sampling not clear. Randomisation procedure not described. | ⊕⊕oo | Use of a hot water bottle with a sterilized wrap on woman's perineum. | Pain severity measured by the McGill pain linear scale | The mean score of pain severity at the second stage of labour showed a significant difference between the two groups (p 0.000) and was lower in the heat therapy group (8.25, SD 1.39) than the routine care group (9.65, SD 1.99) | Heat therapy reduces the labour pain. | |
| Transcutaneous electric nerve stimulation (TENS) | Shahoei et al., 2017a Iran To investigate the effect of transcutaneous electric nerve stimulation on labor pain in 2nd stage | RCT (Women were placed in one of 3 groups) | N = 90 3 groups of 30 nulliparous women; TENS, placebo-TENS, control. | ⊕⊕oo | 3 groups: switched-on TENS, switched-off TENS and control Pain measured on a VAS during 2nd stage | | The severity of pain was lower in the TENS group compared with other groups in 2nd stage of labor (p0.000). No effects on childbirth. | TENS is a safe method for pain relief during childbirth | Very low numbers included, mentioned as limitation in the study. |
| **Pushing techniques** | | | | | | | | | |

(Continued)

**Table 2.** (Continued)

| Midwives' practices | Author, year Country Aim of the study | Study design | Population Group and size (n) (age, parity, ethnicity, etc.) | Quality of study (CASP, Cerqual and GRADE) ⊕⊕⊕⊕ High ⊕⊕⊕o Moderate ⊕⊕oo Low ⊕ooo Very Low | Definitions Main components | Outcomes assessed Effects on outcomes Components associated with outcomes | Results | Key conclusions | Comments |
|---|---|---|---|---|---|---|---|---|---|
| **Pushing/bearing down methods for the second stage of labour** | **Lemos et al., 2017** To evaluate the benefits and possible disadvantages of different kinds of techniques regarding maternal pushing/ breathing during the expulsive stage of labour on maternal and fetal outcomes. | Cochrane Review | 7 Trials (one including women with an epidural: Low et al. 2013) Only presenting results of 6 trials without epidural | ⊕⊕⊕⊕ | Spontaneous versus directed pushing | Duration of second stage, perineal laceration, admission to NNIC, 5 min APGAR score <7, Duration of pushing Spontaneous vaginal delivery | No clear difference in the duration of the 2nd stage of labour (MD 10.26 min (95% CI -1.12 to 21.64 min). No clear difference in 3rd or 4th degree perineal laceration (RR 0.87, 95% CI 0.45 to 1.66), episiotomy (RR 1.05, 95% CI 0.60 to 1.85), duration of pushing (MD -9.76 minutes, 95% CI -19.54 to 0.02) or rate of spontaneous vaginal delivery (RR 1.01, 95% CI 0.97 to 1.05). No difference for neonatal outcomes such as 5' Apgar score <7 (RR 0.35; 95% CI 0.01 to 8.43) and the number of admissions to NICU (RR 1.08; 95% CI 0.30 to 3.79) | No significant difference in the duration of the second stage of labour between spontaneous and directed pushing. Woman's preference and comfort and clinical context should guide decisions. | Only presenting results of 6 trials without epidural (named comparison 1 in the Cochrane review). One trial Low et al. 2013 excluded. |
| **Delayed pushing in lateral position** | **Vaziri et al., 2016** Iran To compare the effects of spontaneous pushing in lateral position with the Valsalva maneuver on maternal and fetal outcomes | RCT (no ITT analyses) | N = 72 randomized; N = 69 analysed. Nulliparous low risk women, live fetus, vertex presentation, 37–40 weeks spontaneous labor. Mean age 22.2 (SD 4.33) | ⊕⊕oo | Intervention: pushing with the urge to push (delayed pushing) in lateral position Control: pushing form beginning of full dilation using Valsalva, supine position | | In intervention group: Less pain severity 7.8 (SD 1.21) vs 9.05 (SD 1.11); p<0.001 Fatigue score 46.59 (SD21) vs 123.36 (SD 43.20); p<0.001 Duration of 2nd stage 76.32 (SD 8.26) vs 64.56 (SD 15.24); p<0.001 Duration of pushing 49.14 (SD11.66) vs 64.56 (SD 15.24); p<0.001 | Spontaneous pushing in the lateral position reduced duration of pushing, fatigue and pain severity, without affecting neonatal outcomes. | |

**Optimising perineal outcomes**

*(Continued)*

**Table 2.** (Continued)

| Midwives' practices | Author, year Country Aim of the study | Study design | Population Group and size (n) (age, parity, ethnicity, etc.) | Quality of study (CASP, Cerqual and GRADE) ⊕⊕⊕⊕ High ⊕⊕⊕o Moderate ⊕⊕oo Low ⊕ooo Very Low | Definitions Main components | Outcomes assessed Effects on outcomes Components associated with outcomes | Results | Key conclusions | Comments |
|---|---|---|---|---|---|---|---|---|---|
| **Perineal techniques for reducing perineal trauma** | **Aasheim et al., 2017** To assess the effect of perineal techniques during the second stage of labour on the incidence and morbidity associated with perineal trauma. | Cochrane review | 22 trials were eligible for inclusion (with 20 trials involving 15,181 women) | ⊕⊕⊕⊕ | Perineal techniques during the second stage of labour | The incidence and morbidity associated with perineal trauma. | Hands on or hands off the perineum made no clear difference in incidence of intact perineum RR 1.03, 95%CI 0.95 to 1.12 (2 studies, 6547 women; moderate-quality evidence), 1st-degree perineal tears RR 1.32, 95% CI 0.99 to 1.77, 2 studies, 700 women; low-quality evidence), 2nd-degree tears (RR 0.77,95% CI 0.47 to 1.28, 2 studies, 700 women; low-quality evidence), or 3rd- or 4th-degree tears (RR 0.68, 95% CI 0.21 to 2.26, 5 studies, 7317 women; very low-quality evidence). Episiotomy was more frequent in the hands-on group (RR 0.58,95% CI 0.43 to0.79, 4 studies, 7247 women; low-quality evidence) A warm compress did not have a clear effect on the incidence of intact perineum (RR 1.02, 95% CI 0.85 to 1.21; 1799 women; 4 studies; moderate-quality evidence), perineal trauma requiring suturing (RR 1.14, 95% CI 0.79 to 1.66; 76 women; 1 study; very low-quality evidence), 1st-degree tears (RR 1.19, 95% CI 0.38 to 3.79; 274 women; 2 studies; very low-quality evidence), 2nd-degree tears (RR 0.95, 95% CI 0.58 to 1.56; 274 women; 2 studies; very low-quality evidence), or episiotomy (RR 0.86, 95% CI 0.60 to 1.23; 1799 women; 4 studies; low-quality evidence). Fewer third- or fourth-degree perineal tears were reported in the warm-compress group (RR 0.46, 95% CI 0.27 to 0.79; 1799 women; 4 studies; moderate-quality evidence). Perineal massage increased the incidence of intact perineum (RR 1.74, 95% CI 1.11 to 2.73, 6 studies, 2618 women; low-quality evidence and substantial heterogeneity between studies). Perineal massage decreased the incidence of 3rd or 4th-degree tears (RR 0.49, 95% CI 0.25 to 0.94, 5 studies, 2477 women; moderate-quality evidence). No clear effect on perineal trauma requiring suturing (RR 1.10, 95% CI 0.75 to 1.61, 1 study, 76 women; very low-quality evidence), 1st-degree tears (RR 1.55, 95% CI 0.79 to 3.05, 5 studies,537 women; very low-quality evidence), or 2nd-degree tears (RR 1.08, 95% CI 0.55 to 2.12, 5 studies, 537 women; very low-quality evidence). Perineal massage may reduce episiotomy (RR 0.55, 95% CI 0.29 to 1.03, 7 studies, 2684 women; very low-quality evidence). One study (66 women) found that women receiving Ritgen's manoeuvre were less likely to have a 1st-degree tear (RR 0.32, 95% CI0.14 to 0.69; very low-quality evidence), more likely to have a 2nd-degree tear (RR 3.25, 95% CI 1.73 to 6.09; very low-quality evidence), and no difference on intact perineum (RR 0.17, 95% CI 0.002 to 1.31; very low-quality evidence). One larger study reported that Ritgen's manoeuvre did not have an effect on incidence of 3rd, or 4th-degree tears (RR 1.24, 95%CI 0.78 to 1.96,1423 women; low-quality evidence). Episiotomy was not clearly different between groups (RR 0.81, 95% CI 0.63 to1.03, two studies, 1489 women; low-quality evidence). Other comparisons: Delivery of posterior versus anterior shoulder first, use of a perineal protection device, different oils/ wax, and cold compresses did not show any effects on outcomes with the exception of increased incidence of intact perineum with the perineal device. Only one study contributed to each of these comparisons. | Moderate-quality evidence suggests that warm compresses, and massage, may reduce third- and fourth-degree tears but the impact of these techniques on other outcomes was unclear or inconsistent. Poor-quality evidence suggests hands-off techniques may reduce episiotomy, but this technique had no clear impact on other outcomes. There were insufficient data to show whether other perineal techniques result in improved outcomes. | For results hands-on hands-off: Substantial heterogeneity for third- or fourth-degree tears means these data should be interpreted with caution. Results massage: Heterogeneity was high for first-degree tear, second-degree tear and for episiotomy—data should be interpreted with caution. |

**Table 2.** (Continued)

| Midwives' practices | Author, year Country Aim of the study | Study design | Population Group and size (n) (age, parity, ethnicity, etc.) | Quality of study (CASP, Cerqual and GRADE) ⊕⊕⊕⊕ High ⊕⊕⊕○ Moderate ⊕⊕○○ Low ⊕○○○ Very Low | Definitions Main components | Outcomes assessed Effects on outcomes Components associated with outcomes | Results | Key conclusions | Comments |
|---|---|---|---|---|---|---|---|---|---|
| Perineal heating pads | Althosseni et al., 2018 Iran To determine the effect of perineal heating pad on the frequency of episiotomies and perineal tears in primiparous females. | Single blind clinical trial | 114 primiparous women recruited, concluding with 54 intervention and 53 control in group Age 18–35 years Singleton, term pregnancies | ⊕⊕○○ | A heated pad was placed on the external region of the perineum. It was placed on the perineum at the start of the second stage of labor, by the trained midwife and removed from the perineum during the mother's transfer to the labour room. | The effect of the perineal heating pad on the frequency of episiotomies and perineal tears. | The results showed a significant difference between the two groups in terms of the episiotomy rate (41% vs 21%, p = 0.025). There was no significant difference between the two groups in terms of frequency of first and second-degree tears, with the first degree tears being observed among 13 (24.1%) and 14 women (26.4%) of the control and intervention groups, respectively. The frequency of second-degree tears in the control and intervention groups was nine (16.7%) and seven (13.2%), respectively. There was no fourth-degree tear in each group. | The results of the current study revealed that the use of perineal heating pad during the second stage of labour can be effective in decreasing the episiotomy rate (statistically significant) and intact perineum (though not statistically significant) in primiparous women. | The results of this study have to be interpreted carefully because of the very low quality of the study. |
| Perineal massage during labour: a systematic review and meta-analysis of randomized controlled trials | Aquino et al., 2018 To evaluate whether perineal massage techniques during vaginal delivery decreases the risk of perineal trauma. | A systematic review and meta-analysis of randomised controlled trials | Nine RCTs reporting on 3,374 women | ⊕⊕⊕○ | Perineal massage during the second stage of labour (with or without the use of water-soluble lubricant) | Primary outcome: Severe perineal trauma. Secondary outcomes were incidences of: episiotomy, first, and second-degree tear and intact perineum. | Women randomised to receive perineal massage during second stage of labour had a significantly lower incidence of severe perineal trauma, compared to those who did not (RR 0.49, 95% CI 0.25–0.94). All the secondary outcomes were not significant, except for the incidence of intact perineum, which was significantly higher in the perineal massage group (RR 1.40, 95% CI 1.01–1.93), and for the incidence of episiotomy which was significantly lower in the perineal massage group (RR 0.56, 95% CI 0.38–0.82). | Perineal massage during labour is associated with significant lower risk of severe perineal trauma, such as third- and fourth-degree tears and episiotomy. Perineal massage was usually done by a midwife in the second stage of labour during or between contractions and during pushing time, with the index or middle finger, using a water-soluble lubricant. | |
| Warm perineal compresses during the second stage of labour for reducing perineal trauma | Magoga et al., 2019 To evaluate the effectiveness of warm compresses during the second stage of labour in reducing perineal trauma. | A systematic review and meta-analysis of randomized controlled trials | Seven trials, including 2103 women | ⊕⊕⊕○ | Women assigned to the intervention group received warm compresses, immersed in warm tap water. These were held against the woman's perineum during and in between pushes in second stage. Warm compresses usually started when the baby's head began to distend the perineum or when there was active fetal descent in the second stage of labour. | The incidence of perineal trauma | A higher rate of intact perineum in the intervention group compared to the control group (22.4% vs 15.4%; RR 1.46, 95% CI 1.22 to 1.74); a lower rate of third degree tears (1.9% vs 5.0%; RR 0.38, 95% CI 0.22 to 0.64), fourth degree tears (0.0% vs 0.9%; RR 0.11, 95% CI 0.01 to 0.86) third and fourth degree tears combined (1.9% vs 5.8%; RR 0.34, 95% CI 0.20 to 0.56) and episiotomy (10.4% vs 17.1%; RR 0.61, 95% CI 0.51 to 0.74). | Warm compresses applied during the second stage of labour increase the incidence of intact perineum and lower the risk of episiotomy and severe perineal trauma. | |
| Hands-on versus hands-off techniques for the prevention of perineal trauma during vaginal delivery | Pierce-Williams et al., 2019 To evaluate whether a hands-on technique during vaginal delivery results in less incidence of perineal trauma than a hands-off technique | A systematic review and meta-analysis of randomized controlled trials | Five RCTs reporting on 7,287 women | ⊕⊕⊕⊕ | Hands-on technique versus hands-off during vaginal delivery | Primary outcome: Severe perineal trauma defined as third- or fourth-degree lacerations. | Women randomized to the hands-on technique had similar incidence of severe perineal trauma (1.5 versus 1.3%; RR 2.00, 95% CI 0.56–7.15). There was no significant between-group difference in the incidence of intact perineum, first-, second- and fourth-degree laceration. Hands-on technique was associated with increased risk of third-degree lacerations (2.6 versus 0.7%; RR 3.41, 95% CI 1.39–8.37) and of episiotomy (13.6 versus 9.8%; RR 1.59, 95% CI 1.14–2.22) compared to the hands-off technique. | Hands-on technique during spontaneous vaginal delivery of singleton gestations results in similar incidence of several perineal traumas compared to a hands-off technique. The incidence of third-degree lacerations and of episiotomy increases with the hands-on technique. | Overall the results are similar to Aasheim et al., (2017) except for the risk of third-degree lacerations. |
| The effect of perineal massage during 2nd stage of labour on multiparous women perineum | Shahoei et al., 2017b Iran To determine the effect of perineal massage in the 2nd stage of labour on perineal lacerations, episiotomy, and perineal pain in multiparous women. | RCT | N = 190 nulliparous women; 38–42 weeks, singleton, vertex position, | ⊕⊕○○ | Perineal massage of 30 min during 2nd stage | Rate of episiotomy and perineal laceration secondary outcomes were comparison of perineal pain after 3 days, 10 days, and 3 months after childbirth. | Episiotomy rate was 69.47% in the intervention group and 92.31% in the control group (p<0.05). The results revealed 23.2% of 1st and 2.1% of 2nd-degree perineal laceration in the intervention group, and no vestibular laceration or 3rd- and 4th-degree lacerations in the intervention group. There were 5.1% of vestibular laceration, 7.7% of 1st-degree laceration, 2.6% of 2nd-degree laceration, and 1.1% of 3rd-degree laceration in the control group. Comparison of postpartum pain showed that the severity of pain 3 days and 3 months after childbirth was significant lower in the intervention group (p = 0.01, p = 0.008, respectively), the severity of pain on the 10th day did not differ significantly (p = 0.78) | Perineal massage during the second stage of labour can reduce the need for episiotomy, and avoid perineal injuries, and perineal pain. | |

NICU: neonatal intensive care unit; MD: mean difference; OR: Odds ratio; RR: relative risk; CI: confidence interval; VAS: visual analogue scale; OASIS: obstetric anal sphincter injuries; TENS: transcutaneous electrical nerve stimulation; SD: standard deviation

Less-experienced midwives were more likely to prefer the "hands off" (72% vs. 41.4%, p<0.001). A higher proportion of midwives in the "hands-off" group would never do an episiotomy (37.1% vs. 24.4%, p = 0.001) for indications other than fetal distress.

A further study explored the views of 31 midwives in the Netherlands, in relation to facilitating women's birthing positions during the second stage of labour [38]. This qualitative study utilised six focus groups to collate the data, which were interpreted using Thachuk's approach [39]. Thachuk's work defines how women are involved in decision making in different maternity care models; for example, the medical model of informed consent in comparison to the midwifery model of informed choice. The influence of midwives' working conditions on the use of birthing positions was an important factor in this study, in particular midwives who conformed to the medical philosophy of care. When asked, 8 (26%) midwives reported that all of the last 10 births they had facilitated was with the woman in the supine position, an additional 6 (19%) midwives stated 8 out of the last 10 were also supine. Midwives suggested that equipment for non-supine births should be more user-friendly. The birth positions midwives preferred were also influenced by their exposure during their initial education and experience during their career. This study acknowledged that giving women informed choice may assist them in using positions that are most appropriate [38].

Begley *et al.*, conducted a focus group study in Ireland and New Zealand among 21 expert midwives to explore techniques used by expert midwives to preserve the perineum intact [37]. In this study a midwife was defined as an "expert" as her practice reflected an episiotomy rate of less than 11.8% (the mean rate from all New Zealand and Irish Midwife-led Unit data combined), rate of women in their care who have an intact perineum of more than 40%, their 'no suture' rate (combination of the number of women with first degree tears that did not require sutures), and a rate of less than 3.2% for serious perineal tears (or one third/fourth degree tear) in the previous 3.5 years of practice. Four core themes were identified: 'Calm, controlled birth', 'Position and techniques in early second stage', 'Hands on or off?' and 'Slow, blow and breathe the baby out.' Using the techniques described enabled these midwives to achieve rates, in nulliparous women, of 3.91% for episiotomy, 59.24% for 'no sutures', and 1.08% for serious lacerations.

## Themes

The remainder of the included studies were primarily intervention studies highlighting evidence-based aspects of midwifery practice during the second stage of labour, with the potential of informing future practice. These empirical findings were synthesised into four main themes namely: birthing positions, non-pharmacological pain relief, pushing techniques and optimising perineal outcomes.

**Birthing positions.** The use of a squatting position is reported to decrease pain severity in the second stage of labour, thus positively affecting labour pain reduction for women. In addition, squatting is viewed as an easy, applicable method to reduce pain during the second stage of labour [32]. Primiparous women who adopt a sitting position are less likely to have an episiotomy and more likely to have a perineal tear [24, 34] with no clear difference however, reported in the number of 3rd or 4th degree perineal tears [24]. It is acknowledged that women should not be discouraged from adopting (semi-)sitting birthing positions to prevent perineal damage. Notably, longer duration of second stage was associated with more women experiencing episiotomies [34]. The upright position is, nonetheless, associated with a reduction in duration of second stage. If progress in labour is slower, then variation in position should be considered, particularly if the woman is in the supine position. Magnetic resonance (MR)

pelvimetry also showed that an upright birthing position significantly expands the female pelvic bony dimensions, suggesting facilitation of labour and birth [34].

**Non-pharmacological pain relief.** Two studies described methods of non-pharmacological pain relief adopted by midwives [29, 30]. A randomised sterilized control trial, using a heat pack (hot water bottle) during the second stage, with a sterilized wrap placed on the woman's perineum for a minimum of five minutes. Pain was assessed using the McGill Pain linear scale during immediately following birth to assess the pain level during the second stage of labour. The mean score of pain severity relating to the second stage of labour showed a statistically significant difference between the two groups (p 0.000) and was lower in the heat therapy group than the routine care group [29]. The effect of transcutaneous electrical nerve stimulation (TENS) on the severity of pain during labour in primiparous women was examined [30]. The findings indicated the severity of pain during the second stage of labour was lower in the TENS group compared with the placebo and control groups ($p = 0.000$).

**Pushing techniques.** During normal physiological birth, when the cervix is fully dilated and/or the fetal head is on the pelvic floor, the mother will feel the urge to push and aided with expulsive contractions maternal pushing will lead to the birth of the baby. In the literature two different techniques of pushing are described: directed, coached, or Valsalva pushing with physiological or spontaneous pushing: Valsalva and physiological or spontaneous pushing. Directed pushing according to the Valsalva technique is repeated, prolonged breath holding and bearing down which causes the glottis to close and increases intrathoracic pressure. Predominantly resulting in closed glottis pushing for 3 to 4 times during each contraction. Physiological or spontaneous pushing is defined as full dilatation of the cervix and commencement of pushing only when women feel the urge to push. No specific instructions are given about timing and duration; mostly resulting in non-directed, multiple short pushes, with no sustained breath holding [25].

Studies comparing these two techniques have been primarily concerned with the effect of pushing style on neonatal acid-base status and/or the length of second stage. Some studies have directly addressed the relationship between the pushing method and perineal or pelvic floor injury or have included it in their analyses. The Cochrane review by Lemos *et al.*, [25] found a mean reduction in the duration of second stage of labour by ten minutes and less third or fourth degree perineal tears, however, these results were not statistically significant and no conclusive (Table 1). A study by Vaziri *et al.*, [33] compared spontaneous pushing with the urge to push (delayed pushing) in lateral position with immediate pushing (from the beginning of full dilation) using Valsalva in supine position. This study concluded that spontaneous pushing in the lateral position reduced duration of pushing, fatigue and pain severity, without affecting neonatal outcomes [33]. While the Cochrane review authors [25] highlighted their inability to report which technique of pushing is best for the mother or baby, the spontaneous pushing technique was found by Vaziri *et al.*, [33] to be a safe method without causing any harm to the baby.

**Optimising perineal outcomes.** There are two main maternity care options to guide the birth of the fetal head, the hands-on or the hands-off (ordinarily with hands-poised) method. The hands-on method aims to prevent severe perineal tears by supporting the perineum during fetal crowning. The other hand is placed on the fetal head and the mother is asked to withhold from pushing, aiming to control the speed of the birth of the head. Lateral flexion of the fetal head is applied to facilitate delivery of the shoulders. With the hands-off (or hands-poised) method the hands do not touch the perineum or fetal head, allowing spontaneous delivery of the head and the shoulders; and the woman is guided in controlled pushing.

A Cochrane review by Aasheim *et al.* [22] found that hands-on or hands-off the perineum showed no clear supporting evidence in the incidence of intact perineum, first degree perineal

tears, second degree tears or third- or fourth-degree tears. However, episiotomy was performed more frequently in the hands-on group. A recent systematic review by Pierce-Williams et al., showed almost similar results. Hands-on technique during spontaneous vaginal delivery of singleton gestations resulted in similar incidence of several perineal traumas compared to a hands-off technique. However, the incidence of third-degree lacerations and of episiotomy increases with the hands-on technique [27].

According to the Cochrane review by Aasheim *et al.* supporting the perineum with a warm cloth or compress did not have a clear effect on the incidence of intact perineum, perineal trauma requiring suturing, first degree tears, second degree tears or episiotomy. However, fewer third or fourth-degree tears were reported in the warm-compress group [22]. A recent systematic review of Magoga et al., however, showed that warm compresses applied during the second stage of labour increases the incidence of intact perineum and lower the risk of episiotomy and severe perineal trauma. This systematic review included seven trials reporting on 2,103 women. This study showed that the use of a perineal heating pad during the second stage of labour can be effective in decreasing the episiotomy rate in primiparous women [26]. These results are consistent with the study of Alihosseni *et al.* [28].

Perineal massage during labour is usually done in the second stage, during or between contractions and during pushing time, with the index and middle fingers, using a water-soluble lubricant. The purpose of this technique is to gently stretch the perineum from side to side. Perineal massage increased the incidence of intact perineum and decreased the incidence of third- or fourth-degree tears. Perineal massage had no clear effect on first or second degree suturing, however, it may reduce episiotomy [22] A further study examined the effectiveness of perineal massage [31] showing that in primiparous women a perineal massage of 30 minutes during the second stage of labour reduced the episiotomy rate (69% in the massage group, and 92% in the control group). According to a recent systematic review and meta-analysis of nine randomised controlled trials reporting on 3374 women, perineal massage during second stage of labour is associated with significant lower risk of severe perineal trauma, such as third- and fourth-degree lacerations and episiotomies [23].

Additional findings relating to other midwifery practices during the second stage of labour were also reported within the Cochrane review [33], including: whether the posterior or the anterior shoulder should be born first, the use of different oils/wax or cold compress on the perineum and the use of a perineal protection device. For the majority it is not clear if these techniques had a beneficial effect on preventing perineal trauma, with the exception of an increased incidence of intact perineum with the use of a perineal protection device.

## Discussion

This systematic review focused specifically on midwives' practices during the second stage of labour for women experiencing a physiological labour and birth. The results provide insight in how midwives practices are influenced by their years of experience, the designation of the maternity unit where they work, (for example, a midwife-led unit or an obstetric unit) and that midwives practices are not always consistent with the scientific literature or with a physiological approach to birth.

In relation to birthing positions, women can adopt various positions to give birth, largely, upright (such as, standing, squatting, kneeling) and supine (such as lateral, lithotomy, dorsal, semi-recumbent). The limited number of studies relating to birth position included in this review reported on perineal damage and pain severity and included midwives' perspectives/practices. Ultimately, women should be facilitated to adopt the position they deem most

comfortable to give birth and should be educated with regards to all childbirth positions, encouraging them to select each of the positions voluntarily.

For non-pharmacological pain relief, transcutaneous electrical nerve stimulation seems to be effective in reducing pain during birth and it has no consequences for women and their infants [30]. The empirical evidence also supports the use of heat therapy in the form of a heat pack for women in the physiological second stage of labour, as it can effectively reduce labour pain [29]. No included studies discussed the effects of water on reducing pain during birth.

Regarding pushing techniques, we found no significant difference in the duration of the second stage of labour between spontaneous and directed pushing. While a Cochrane review highlighted an inability to report which technique of pushing is best for the mother or baby. Woman' preference, comfort and clinical context should therefore guide decisions [25].

As highlighted above a Cochrane review [22] and a systematic review by Pierce-Williams et al. [27] found that hands-on or hands-off the perineum showed no clear supporting evidence in the incidence of intact perineum, first degree perineal tears, second degree tears or fourth degree tears, with episiotomy being performed more frequently in the hands-on group. These reviews were inconsistent regarding third degree tears. The lack of heterogeneity of studies within the Cochrane review for third-or fourth-degree tears means these data should be interpreted with caution. In conclusion, there is insufficient evidence to promote one of these midwifery practices over the other in regard to preventing perineal tears [22].

High-quality evidence suggests that compresses emerged in warm tap water increase the incidence of intact perineum and lower the risk of episiotomy and third and fourth-degree tears [26]. This low-cost highly effective intervention could easily be implemented in all birth settings. To optimise perineal outcomes during the second stage of labour, perineal massage can reduce the need for episiotomy, avoid perineal injuries and perineal pain [22].

## Strengths and limitations

This is a full systematic review with searches across multiple databases reporting on published research on how midwives can facilitate women to give birth during the physiological second stage of labour. The methods of our review are transparent with full protocol published in PROSPERO in advance of the review [40].

In view of the variable risk of bias of the included trials, further trials using well-designed protocols are needed to ascertain the true benefits and risks of various midwifery practices during the second stage of labour.

When studying research about how to facilitate women to give birth during physiological second stage of labour, we came upon scarce evidence regarding the care and support provided by midwives. These non-clinical aspects of labour and birth matter to woman, and are essential components of quality intrapartum care for women and their family [WHO Intrapartum care 2018]. Only one article was included in our systematic review regarding this [37]. Begley et al. underlined in her qualitative study the importance of developing an empowering, trusting relationship with the woman, ensuring a quiet, calm environment, reassuring and supporting the woman to optimise her birth outcome. There is a dearth of evidence relating to non-clinical aspects of midwives' practice during the second stage of labour, such as continuous support, emotional support, companionship, effective communication and respectful care. These aspects of care are often not regarded as priorities [7]. Perhaps this is because not all midwives' practices are documented and therefore researched. More research is needed on how midwives practices may affect a woman's experience of labour and birth outcomes.

For this review the second stage of labour was defined as the time period between full dilatation of the cervix and the birth of the baby, whilst the woman is experiencing an involuntary

urge to bear down, due to expulsive uterine contractions [7]. However, another definition of the second stage of labour has been noted. Bjelke et al. outlines a definition of the second stage of labour, which included two phases, the passive and the active phase [41]. The passive phase is defined as full dilatation of the cervix before or in the absence of involuntary expulsive contractions. During this phase the presenting part descends passively down in the maternal pelvis, eventually generating a reflex that causes a strong urge to push. The active phase is the stage of expulsive efforts. This division of the second stage of labour, into two phases is rarely reported. Further research could focus on how to manage the passive phase of the second stage of labour.

Culture, birth settings and work practices effect the possibility of the physiological approach to birth being enabled or not [35]. It is essential therefore that women with a straightforward pregnancy* [42] can take an informed choice [43] and gain access to midwife-led services to plan their birth at home or within a midwife-led unit, where the physiological approach to birth is enabled. Gaining access to a midwife-led unit can be enabled by utilising an evidenced-based guideline for admission to either an alongside or freestanding midwife-led unit and midwives can facilitate care by following a normal labour and birth care pathway [42, 44].

## Conclusion

This review systematically collated pertinent literature by retrieving 6,382 studies after the removal of duplicates. Following synthesis empirical evidence of different aspects of midwifery practices relating to care during the second stage of labour were retrieved including: Birthing positions, non-pharmacological pain relief, pushing techniques and optimising perineal outcomes. By implementing this evidence midwives may enable women during the second stage of labour to optimise physiological processes to give birth. There is however, a dearth of evidence relating to midwives' practice during the second stage of labour and further robust studies are required. There is also limited knowledge of how midwives' practices may affect a woman's experience of the second stage of labour. Nevertheless, this systematic review provides a summary of the current empirical evidence of midwives' practices of physiological second stage of labour and can inform midwifery practice, education and future research in the support of high-quality intrapartum care.

*Straightforward singleton pregnancy, is one in which the woman does not have any pre-existing condition impacting on her pregnancy, a recurrent complication of pregnancy or a complication in this pregnancy which would require on-going consultant input, has reached 37 weeks' gestation and $\leq$ Term +14 days [42].

## Supporting information

**S1 Checklist. PRISMA 2009 checklist.**
(DOC)

**S1 Table. Search strategy tables.**
(DOCX)

## Acknowledgments

The authors gratefully thank Mary Dharmachandran (subject librarian at the Royal College of Midwives, UK) for her valuable contribution to this systematic review.

## Author Contributions

**Conceptualization:** Maria Healy, Corine J. Verhoeven.

**Data curation:** Maria Healy, Viola Nyman, Dale Spence, René H. J. Otten, Corine J. Verhoeven.

**Formal analysis:** Maria Healy, Dale Spence, Corine J. Verhoeven.

**Funding acquisition:** Maria Healy.

**Investigation:** Viola Nyman, Corine J. Verhoeven.

**Methodology:** Maria Healy, René H. J. Otten, Corine J. Verhoeven.

**Project administration:** Maria Healy.

**Software:** Maria Healy.

**Supervision:** Maria Healy.

**Writing – original draft:** Maria Healy, Viola Nyman, Dale Spence, René H. J. Otten, Corine J. Verhoeven.

**Writing – review & editing:** Maria Healy, Viola Nyman, Dale Spence, René H. J. Otten, Corine J. Verhoeven.

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
