## [Decision Letter · Decision Letter 0]

3 Sep 2019

PONE-D-19-16084

How do midwives facilitate women to give birth during physiological second stage of labour? A systematic review

PLOS ONE

Dear Dr Healy,

Thank you for submitting your manuscript to PLOS ONE. After careful consideration, we feel that it has merit but does not fully meet PLOS ONE’s publication criteria as it currently stands. Therefore, we invite you to submit a revised version of the manuscript that addresses the points raised during the review process.

This is an excellent manuscript, although you will note that the editorial office has recommended some restructuring. The points raised by the reviewers are minor and will strengthen the overall manuscript.

We would appreciate receiving your revised manuscript by Oct 18 2019 11:59PM. To enhance the reproducibility of your results, we recommend that if applicable you deposit your laboratory protocols in protocols.io, where a protocol can be assigned its own identifier (DOI) such that it can be cited independently in the future. For instructions see: http://journals.plos.org/plosone/s/submission-guidelines#loc-laboratory-protocols

We look forward to receiving your revised manuscript.

Kind regards,

Christine E East

Academic Editor

PLOS ONE

Journal Requirements:

2. Please include your tables as part of your main manuscript and remove the individual files. Please note that supplementary tables (should remain/ be uploaded) as separate "supporting information" files

3. We note that your article has been submitted as a "Collection Review" article type, but is research submitted to the BIRTH Collection. When resubmitting your manuscript, we ask that you update your article type to "Research Article" in the online submission form. Please note that some fields in the submission form, particularly in the "Additional Information" field, will have been reset with this change, so please go through your submission in full to ensure that all information is accurate and complete when resubmitting your manuscript.

4. We note that the research questions of your systematic review are very broad and the result and discussion consist of a narrative synthesis. In light of the methodological approach, used we feel that a scoping review may be a more suitable article type for your study. Therefore, we ask you if could please consider replacing the PRISMA checklist with the newly developed scoping review extension (PRISMA-ScR; available at http://www.equator-network.org/reporting-guidelines/prisma-scr/), making any relevant changes to the main manuscript.

In addition, we request to address the following points:

-    Please discuss the results of the quality assessment in the Results section

-    Please update the search of your systematic review, including all new studies published since May 2018

Thank you for your attention to these requests.

Information for the Academic Editor (as provided by the journal staff)

Note from Staff Editor Dario Ummarino (dummarino@plos.org):During our initial assessment of this manuscript we noted that a scoping review methodology would be more appropriate for this article. Therefore, we will ask the authors to address the following editorial requests, which will be added to the letter at the time of first decision:""We note that the research questions of your systematic review are very broad and the result and discussion consist of a narrative synthesis. In light of the methodological approach, used we feel that a scoping review may be a more suitable article type for your study. Therefore, we ask you if could please consider replacing the PRISMA checklist with the newly developed scoping review extension (PRISMA-ScR; available at http://www.equator-network.org/reporting-guidelines/prisma-scr/), making any relevant changes to the main manuscript.In addition, we request to address the following points:-Please discuss the results of the quality assessment in the Results section-Please update the search of your systematic review, including all new studies published since May 2018.""For your information, a scoping review (also scoping study) refers to a rapid gathering of literature in a given policy or clinical area where the aims are to accumulate as much evidence as possible and map the results. Scoping reviews are a type of literature review that aims to provide an overview of the type, extent and quantity of research available on a given topic. By â€˜mappingâ€™ existing research, a scoping review can identify potential research gaps and future research needs, and do so by using systematic and transparent methods. For guidance in the assessment of this scoping review, you might refer to a newly developed extension of the PRISMA checklist (PRISMA-ScR), which is available at http://www.equator-network.org/reporting-guidelines/prisma-scr/ you have any question on the above or want to discuss this manuscript further please do not hesitate to contact me.

Reviewer #1: Thank you for the opportunity to review your article. It was a joy to read such a high quality paper with a very interesting topic. The discussion part is the weakest part of your paper and I would suggest to work on this to get a stronger statement, you have begun in the introduction so strong and clear and this is in my opinion lost in the discussion.

Abstract

Here you are writing that you conducting "high quality intrapartum care" for the reader it would be helpful to read that it is about the second stage of childbirth.

Introduction

- Second sentence: I would suggest to cite a publication like this to have also a reference mentioned with physical effects of childbirth, even that your focus is another, but you have here a brought statement:

Keag OE, Norman JE, Stock SJ (2018) Long-term risks and benefits associated with cesarean delivery for mother, baby, and subsequent pregnancies: Systematic review and meta-analysis. PLoS Med 15(1): e1002494. https:// doi.org/10.1371/journal.pmed.1002494

- You write that the first stage of labour is characterised by regular "painful" contractions. Pain is something subjective. Not every woman is experiencing it as "painful" like woman who practicing hypnobirthing are coping quite well, in my experience. Please consider if you take the word out to have a more neutral definition.

- The definition/reference that you used for normal physiological birth is also saying that interventions should try to be avoided. You mention that normal physiological birth is associated with non-use of epidural, what is with other pharmacological pain-relief? As a reader I do not know how I should interpret your statement, here.

- I would also advice to rite something about augmentation, because you exclude it in your search but in the definition it is not clear that it is not part of a normal birth.

- I like the following section very much with the hormones!

- What do you mean with "good quality intrapartum care" in your second RQ? Through your paper it is not getting clear. The other research question is addressed very well in the text!

Method

-Fig. 2 is perfectly clear, but in the text it is not clear where you combined the AND for the second stage and the NOT for animals and humans.

-The (S1 tables) is a bit lost here, and it would be helpful to be clearer what you mean.

- I can only guess what you want to say with the statement "reflecting the NICE intrapartum care guidance". I would love to read here your reasons for deciding on using NICE in an EU project, where a lot of other countries have also evidence-based guidelines that as you mentioned would be no problem to translate based on the COST members.

Results

-You mention Barasinski et al, but there is no number, as a reader I must check the ref list to know if you already mentioned it as number 21 or 22. Can you please add the number?

- It would be helpful for the structure to have a sub-heading after the first paragraph before you go on to the remainder studies that you have synthesised in themes.

- p. 11on "Perineal massage" here it is not explicied if the massage is ment during pregnancy or in the second stage.

- You write on p. 8 that the evidence-based aspect of midwifery practice during the second stage …. . I was wondering what is your comparison basis for "evidence-based" in this sentence. In your Method section, as I have read it, it is not defined. On page 12 in the discussion you make also a statement on the midwifery practice that "is not always evidence-based.".

Discussion

-Here the first research question "how do midwives facilitate women to give birth during …" could be more discussed.

-I would have enjoyed to read more about the research gaps that you could not answer because of missing research in relation to your tow RQ. I do not found it enough to read it only in the conclusion. What are the next steps?

In the introduction you have cite the WHO 2018 on satisfaction, why do you mention this aspect not again in the discussion part?

Tables and Figures

They are all very good, thank you!

Reviewer #2: Dear authors,

I had a privilege to read your manuscript, which is well constructed and well written. In general, the title of the article is concise and reflects the content. The logical flow of the content is also clear as well as the scientific rigour of systematic review that you followed correctly. Despite this excellent manuscript, I would suggest that you delete the subtitle in the Discussion section that is obsolete ("Principal Findings"). In my opinion, "Discussion" section is the weakest part of your manuscript and needs some more work. I see a great deal of issues in the article that raise the question of professionalism of midwifery, autonomy, implementation of midwifery model of care in clinical practice etc. I think this would appeal international readers to read your article even more. Please consider this.

Sincerely,

MP
---

## [Author Response · Author response to Decision Letter 0]

17 Oct 2019

Cover letter minor revision

Christine E East,

Academic Editor

PLOS ONE

Dr Maria Healy

Lecturer in Midwifery (Education)

School of Nursing and Midwifery

Queen's University Belfast

97 Lisburn Road

Belfast, Northern Ireland

BT9 7BL

Telephone: +44 28 9097 2394

Email: Maria.Healy@qub.ac.uk

PONE-D-19-16084

How do midwives facilitate women to give birth during physiological second stage of labour? A systematic review.

Maria Healy, Ph.D.; Viola Nyman, Ph.D.; Dale Spence, Ph.D; Rene H Otten; Corine Verhoeven, Ph.D.

Belfast, 18 October 2019

Dear Editor, Christine East, 

We thank you and the reviewers for their comments on our manuscript entitled “How do midwives facilitate women to give birth during physiological second stage of labour? A systematic review” which we submitted for publication in PLOS ONE and for giving us the opportunity to revise the manuscript taking the comments into account.

We read the comments with interest and we adjusted the manuscript accordingly. We copied the comments of the reviewers and wrote our response in Italics underneath each comment. We included a revised manuscript with track changes and a clean version of the revised manuscript. 

We hope these adjustments make the manuscript suitable for publication in your journal and we are looking forward to your reply. 

Yours sincerely, on behalf of the co-authors,

Dr Maria Healy

Response to the comments of the editor and reviewers. 

We note that the research questions of your systematic review are very broad, and the result and discussion consist of a narrative synthesis. In light of the methodological approach, used we feel that a scoping review may be a more suitable article type for your study. Therefore, we ask you if could please consider replacing the PRISMA checklist with the newly developed scoping review extension (PRISMA-ScR; available at http://www.equator-network.org/reporting-guidelines/prisma-scr/), making any relevant changes to the main manuscript.

Thank you for this suggestion. However, we feel it would be appropriate to keep it as a systematic review. Our reasons for this are as following: 

1. The protocol has been published stating we are undertaking a systematic review

2. All Systematic Review steps have been followed in detail 

3. Systematic reviews do frequently use a narrative synthesis – rather than a meta-analysis 

4. The questions were broad to capture a sufficient data base of studies to make it a meaningful systematic review.

However, we are willing to compromise to call our paper a systematic integrative review, if required by the editor.

Please discuss the results of the quality assessment in the Results section

These were added in the Results section, see page 8.

Please update the search of your systematic review, including all new studies published since May 2018.

Further to the editors request, we updated the search of our systematic review, including all new studies published since May 2018 until 5th September 2019 and have adjusted the Results and the Discussion section accordingly. 

Please include captions for your Supporting Information files at the end of your manuscript, and update any in-text citations to match accordingly.

We added captions for the Supporting Information files at the end of our manuscript.

Reviewer #1: 

The discussion part is the weakest part of your paper and I would suggest to work on this to get a stronger statement, you have begun in the introduction so strong and clear and this is in my opinion lost in the discussion.

Thank you for this comment. We have further developed the Discussion section.

Abstract

Here you are writing that you conducting "high quality intrapartum care" for the reader it would be helpful to read that it is about the second stage of childbirth.

We added this in our revised manuscript.

Introduction

Second sentence: I would suggest to cite a publication like this to have also a reference mentioned with physical effects of childbirth, even that your focus is another, but you have here a brought statement:

Keag OE, Norman JE, Stock SJ (2018) Long-term risks and benefits associated with cesarean delivery for mother, baby, and subsequent pregnancies: Systematic review and meta-analysis. PLoS Med 15(1): e1002494. https:// doi.org/10.1371/journal.pmed.1002494

Thank you for this suggestion, we added a sentence in the Introduction and included this reference.

You write that the first stage of labour is characterised by regular "painful" contractions. Pain is something subjective. Not every woman is experiencing it as "painful" like woman who practicing hypnobirthing are coping quite well, in my experience. Please consider if you take the word out to have a more neutral definition.

We do agree that women who practice hypnobirthing experience contractions in the first stage of labour not as painful however as most women will experience regular painful contractions, and as ‘painful’ is also stated in the WHO Intrapartum Care for a Positive Childbirth Experience Guideline first stage of labour definition (page 3); no adjustments were made in the revised manuscript.

The definition/reference that you used for normal physiological birth is also saying that interventions should try to be avoided. You mention that normal physiological birth is associated with non-use of epidural, what is with other pharmacological pain-relief? As a reader I do not know how I should interpret your statement, here.

We added ‘or other pharmacological pain relief’ to this sentence.

I would also advice to write something about augmentation, because you exclude it in your search but in the definition, it is not clear that it is not part of a normal birth.

We added a sentence about augmented and induced labour.

I like the following section very much with the hormones!

Thank you.

What do you mean with "good quality intrapartum care" in your second RQ? Through your paper it is not getting clear. The other research question is addressed very well in the text!

We removed this question since it is more or less the same question as our main research question, only phrased differently.

Method

Fig. 2 is perfectly clear, but in the text it is not clear where you combined the AND for the second stage and the NOT for animals and humans.

This information is provided in Supporting Information S2 Table. Search strategy tables

The (S1 tables) is a bit lost here, and it would be helpful to be clearer what you mean.

Here we meant Supporting Information S2 Table. Search strategy tables. We wrote this in revised manuscript.

I can only guess what you want to say with the statement "reflecting the NICE intrapartum care guidance". I would love to read here your reasons for deciding on using NICE in an EU project, where a lot of other countries have also evidence-based guidelines that as you mentioned would be no problem to translate based on the COST members.

The NICE Intrapartum care guidance is an internationally well-known and highly valued guideline, used or referred to in many countries.

Results

You mention Barasinski et al, but there is no number, as a reader I must check the ref list to know if you already mentioned it as number 21 or 22. Can you please add the number?

We apologize for this. We added the corresponding reference number in the manuscript.

It would be helpful for the structure to have a sub-heading after the first paragraph before you go on to the remainder studies that you have synthesised in themes.

We added the subheading “Themes” to the manuscript.

p.11 on "Perineal massage" here it is not explicated if the massage is meant during pregnancy or in the second stage.

Since our review focusses on what midwives can do to facilitate giving birth during the second stage of labour, perineal massage is meant during the second stage of labour. We clarified this in the revised manuscript.

You write on p. 8 that the evidence-based aspect of midwifery practice during the second stage …. . I was wondering what is your comparison basis for "evidence-based" in this sentence. In your Method section, as I have read it, it is not defined. On page 12 in the discussion you make also a statement on the midwifery practice that "is not always evidence-based.".

We changed “evidence-based into “consistent with the scientific literature”. See pages 8 and 12.

Discussion

Here the first research question "how do midwives facilitate women to give birth during …" could be more discussed.

Thank you for your comment. We further enhanced the Discussion section.

I would have enjoyed to read more about the research gaps that you could not answer because of missing research in relation to your tow RQ. I do not find it enough to read it only in the conclusion. What are the next steps?

We added a paragraph regarding research gaps and next steps.

In the introduction you have cite the WHO 2018 on satisfaction, why do you mention this aspect not again in the discussion part?

We mentioned the WHO 2018 in the Discussion section, as suggested by the reviewer.

Reviewer #2:

I would suggest that you delete the subtitle in the Discussion section that is obsolete ("Principal Findings"). 

Following the suggestion of the reviewer, we deleted the subtitle in the Discussion section. (page 14)

In my opinion, "Discussion" section is the weakest part of your manuscript and needs some more work. I see a great deal of issues in the article that raise the question of professionalism of midwifery, autonomy, implementation of midwifery model of care in clinical practice etc. I think this would appeal international readers to read your article even more. Please consider this.

Thank you for this addition. We rewrote the Discussion section.

---

## [Editor Report · Decision Letter 1]

2 Dec 2019

How do midwives facilitate women to give birth during physiological second stage of labour? A systematic review

PONE-D-19-16084R1

Dear Dr. Healy,

We are pleased to inform you that your manuscript has been judged scientifically suitable for publication and will be formally accepted for publication once it complies with all outstanding technical requirements.

With kind regards,

Christine E East

Academic Editor

PLOS ONE
---

## [Editor Report · Acceptance letter]

8 Jul 2020

PONE-D-19-16084R1 

How do midwives facilitate women to give birth during physiological second stage of labour? A systematic review 

Dear Dr. Healy:

I'm pleased to inform you that your manuscript has been deemed suitable for publication in PLOS ONE. Congratulations! Your manuscript is now with our production department. 

Kind regards, 

on behalf of

Dr. Christine E East 

Academic Editor

PLOS ONE